# Nanotechnology-Based Combinatorial Anti-Glioblastoma Therapies: Moving from Terminal to Treatable

**DOI:** 10.3390/pharmaceutics14081697

**Published:** 2022-08-15

**Authors:** Amir Barzegar Behrooz, Zahra Talaie, Amir Syahir

**Affiliations:** 1Nanobiotechnology Research Group, Department of Biochemistry, Faculty of Biotechnology and Biomolecular Science, Universiti Putra Malaysia, Serdang 43400, Malaysia; 2School of Biology, Nour Danesh Institute of Higher Education, Isfahan 84156-83111, Iran; 3UPM-MAKNA Cancer Research Laboratory, Institute of Bioscience, Universiti Putra Malaysia, Serdang 43400, Malaysia

**Keywords:** glioblastoma, nanotechnology, nanomedicine, nanocarriers, drug delivery

## Abstract

Aggressive glioblastoma (GBM) has no known treatment as a primary brain tumor. Since the cancer is so heterogeneous, an immunosuppressive tumor microenvironment (TME) exists, and the blood–brain barrier (BBB) prevents chemotherapeutic chemicals from reaching the central nervous system (CNS), therapeutic success for GBM has been restricted. Drug delivery based on nanocarriers and nanotechnology has the potential to be a handy tool in the continuing effort to combat the challenges of treating GBM. There are various new therapies being tested to extend survival time. Maximizing therapeutic effectiveness necessitates using many treatment modalities at once. In the fight against GBM, combination treatments outperform individual ones. Combination therapies may be enhanced by using nanotechnology-based delivery techniques. Nano-chemotherapy, nano-chemotherapy–radiation, nano-chemotherapy–phototherapy, and nano-chemotherapy–immunotherapy for GBM are the focus of the current review to shed light on the current status of innovative designs.

## 1. Introduction

In the central nervous system (CNS), glioblastoma multiforme (GBM) accounts for almost half of all primary malignant tumors [1]. If surgical resection is not feasible, the best treatment option for GBM is chemotherapy and/or radiation. Because of the limited and ineffective GBM treatment choices, new procedures and improvements to current technology are urgently required [1]. Nanotechnology has provided excellent therapy options for various diseases over the last several decades because of its capacity to increase drug systemic administration and absorption [2,3]. Due to the complicated mechanisms at play in the development, progression, and invasion processes of GBM, increasing evidence suggests that a single treatment method may lead to drug resistance and tolerance by cancer cells, leading to metastasis, as well as recurrence [4,5]. Because of this, the use of many treatment agents with different modes of action should be able to overcome these issues. Short half-life in circulation, poor delivery to the diseased locations, and difficulties controlling the release of different drugs at the correct places are significant issues with the present treatments. These issues result in a lack of therapeutic drug accumulation in the tumor cells, preventing proper tumor elimination [6,7].

In addition, because of the tumor’s location, the BBB is a significant and crucial barrier to drug delivery [8,9]. The BBB prevents the passage of almost all small-molecule medications and nearly all large molecular pharmaceuticals [10,11]. GBM therapy drug delivery obstacles include extended blood circulation, adequate transportation across the BBB, efficient internalization, and regulated drug release inside the GBM cells. In order to guarantee adequate therapeutic agents that accumulate in GBM cells, all of these processes must be taken into consideration. Nanotechnology has been used in GBM therapies to bypass these physiological limitations and increase therapeutic efficacy. Few clinical trials employing nano-therapies for GBM therapy have been published, despite many in vivo and in vitro studies proving the efficacy and therapeutic potential of nanotechnology and/or nanocarriers [12,13,14,15]. Gene therapy, immunotherapy, phototherapy, and thermotherapy are further anti-glioma treatment modalities that have been used in addition to the usual therapy [16,17,18,19,20]. In this review, a variety of anti-glioblastoma combined therapy techniques are discussed.

## 2. Biological Challenges of Glioblastoma Therapy: More Than Meets the Eye

### 2.1. Challenge 1. Heterogeneity as a Big Challenge Ahead in Targeting Glioblastoma

GBM is still incurable, and survival rates have only modestly improved despite improvements in treatment results over the last several decades. The discovery of tumor heterogeneity can partly explain the ineffectiveness of existing antiproliferative therapies. Resistance to combination oncology is secured by dynamic heterogeneity. Therapy-resistant subclones form while tumor growth continues. As reflected in its name, ‘multiforme’, heterogeneity is a well-known characteristic of GBM. In the early stages of mitosis, many subclones are formed. It becomes more difficult to determine the true genetic condition as the process progresses [21,22]. Cellular and molecular heterogeneity is seen across cancers and within the same tumor itself, which is remarkable [23,24]. There have been multiple genetic studies proving the presence of distinct cell types, which shows that GBM may emerge from diverse cells. Glioblastoma is most likely a group of disorders, as demonstrated by the wide range of genetic variants found in the disease. Where does glioblastoma get its wide range of subtypes, however? The solution to this issue might be found in the cell origin of GBM. The classical, mesenchymal, neural, and proneural subtypes of GBM have been identified by expression profiling of glioblastoma samples [25]. Malignant gliomas may develop in any part of the central nervous system (CNS). Proneural and neural gliomas develop in or near the subventricular zone, whereas mesenchymal and classical gliomas develop away from the subventricular zone. The dedifferentiation and stem-cell theories are the two leading hypotheses for glioblastoma’s biological origin. The “dedifferentiation hypothesis or stochastic model” claims that all cells are equal; however, under different genetic or epigenetic inputs, only some of them can promote tumor development. The “hierarchical model or stem-cell hypothesis” states that cancers include a fraction of cells, known as cancer stem cells (CSCs), capable of proliferating, giving birth to, and reseeding a tumor. Both of these scenarios seem reasonable in light of the available data [25]. Treatment efficacy may be improved by combining therapies that target numerous subclones simultaneously and use multiple parallel pathways (Figure 1). 

### 2.2. Challenge 2. Tumor Microenvironment

In GBM, treatment resistance and tumor recurrence may be linked to the tumor microenvironment (TME) [26]. Diverse cell types are seen in the TME, e.g., tissue-resident cells such as neurons and astrocytes, myeloid cells such as the resident microglia, bone marrow-derived macrophages (BMDMs), bone marrow-derived DCs, and neutrophils, lymphoid cells, endothelial cells, and pericyte/fibroblast-derived cells. The extracellular matrix (ECM) encompasses all of these cells [27,28,29,30]. Both normal tissue homeostasis and tumor progression need a bidirectional cell–microenvironment interaction. Cancer cells in particular interact with the surrounding stroma and affect disease start, development, and patient outcomes [31]. Tumor cells, microglia, and tumor-associated macrophages (TAMs) secrete numerous cytokines, which cause an immunosuppressive state in the TME of GBM [29]. Cells in the GBM microenvironment release several cytokines and chemokines, as well as growth factors, extracellular vesicles, and proteins in order to create a favorable microenvironment. Tumor immune evasion is facilitated by the interactions of TME cells with one other and with neoplastic cells through suppressor receptors such as PD-1, CTLA-4, CD70, and gangliosides. Inhibition of immune response and activation of FoxP3^+^ regulatory T-cells (Tregs) are two of the most important outcomes of this process, as are suppression of NK activity, T-cell activation, induction of T-cell apoptosis, and downregulation of MHC expression [26,30,32,33] (Figure 2). 

### 2.3. Challenge 3. Blood–Brain Barrier (BBB) vs. Blood–Tumor Barrier (BTB)

The basement membrane, pericytes, and perivascular astrocyte end-foot processes make up the majority of the BBB, which is primarily made up of a layer of non-fenestrated capillary endothelial cells covered in glycocalyx and connected via a web of intercellular tight junctions (TJs) and adherens junctions. There are three layers of the BBB: the glycocalyx, endothelial layer, and extravascular layer [34]. The integrity of the BBB is disturbed in both primary and metastatic brain tumors, resulting in the formation of the so-called brain–tumor barrier (BTB). The BTB is accompanied by reduction in the expression of TJs and in the secretion of vascular endothelial growth factor (VEGF) from tumor cells, an increase in the number of reactive astrocytes, shrinking of astrocyte end-feet, and a breakdown of the basal membranous membrane. In glioma, the BBB is disrupted in a variety of ways, depending on the stage of the illness. Because of the increased VEGF expression and angiogenesis inside hypoxic zones, with a more immature and permeable vasculature within the tumor, it corresponds with a greater grade of malignancy. When the BBB is intact, TJs prevent big molecules from passing through the interendothelial slits and pores; however, when the BBB is damaged or the BTB is present, TJs are absent, allowing large molecules to pass through. A leaky BBB/BTB implies that the BBB is no longer restricting drug delivery and effectiveness in treating glioblastoma treatments (GBM). A growing body of information suggests, however, that the heterogeneous breakdown of the BBB in GBM makes it impossible to achieve uniform drug concentrations inside the tumor. As a result, novel drug delivery methods to the brain should be developed to avoid an intact BBB/BTB [35,36,37,38,39] (Figure 3).

## 3. Multimodality Therapeutic Approaches in Glioblastoma

### 3.1. Current Treatment 

The mainstay of treatment for GBM patients comprises a maximal amount of surgical resection followed by chemotherapy and/or radiation, if that is practicable. In the treatment of GBM, the current gold standard is TMZ. Alkylating chemical TMZ promotes apoptosis by methylating DNA’s purines. Because of O6-methylguanine-DNA methyltransferase expression, TMZ does not work. By causing double-strand breaks in DNA, radiation treatment triggers tumor cell death. However, radiotherapy for brain tumors may also cause tumor recurrence or secondary gliomas [41,42,43]. New treatments for GBM are urgently required to increase treatment effectiveness and target GBM tumor cells because of the present standard of care’s shortcomings. Synergistic therapeutic effects may be achieved without additional toxicity, improved effectiveness, or overcoming medication resistance in cancer patients via the use of combination therapy. In addition, a wide range of benefits such as drug payloads, longer blood circulation, lower dose frequency necessary for therapeutic effectiveness, and uniform and sustained drug release kinetics are included in combination treatment employing nanoparticles and anticancer drugs [44].

### 3.2. Nanotechnology as the Potential Therapeutic Strategy for Drug Delivery to Glioblastoma 

Nanoparticles (NPs) have been used in a wide variety of therapeutic settings in recent years. Systemic, microenvironmental, and cellular barriers that differ across patients and diseases have been traversed by NPs, which were created to circumvent the limits of free therapies. These biological hurdles to delivery have been overcome by more modern NP designs that combine complex structures, bio-responsive moieties, and targeting agents into their design. They may, thus, be used in increasingly sophisticated systems, including nanocarrier-mediated combinations, to modify several pathways and enhance therapeutic effectiveness against specific macromolecules, target certain stages of the cell cycle, or overcome mechanisms of drug resistance [45]. It is possible that the use of a nanocarrier for cancer treatment might enhance therapeutic effectiveness and safety by protecting the medication from degradation, improving solubility, extending plasma half-life, and increasing tumor accumulation while also allowing for continuous drug release. The increased permeability and retention (EPR) effect allows nanocarriers to passively target solid tumors. Nanocarrier extravasation and aggregation in the tumor site are made possible by the specific physicochemical properties of the tumor’s leaky vasculature and poor lymphatic drainage [46,47,48,49] (Figure 4). 

Fast renal clearance is a problem for nanostructures less than 10 nm in size. Structures larger than 200 nm may be filtered out of the bloodstream by the liver and recognized by the reticuloendothelial system (RES). Hydrophobicity, surface charge, and antiaggregating properties are all critical in preventing opsonization, immunogenicity, and other undesirable effects. Cationic NPs are more likely to be absorbed than anionic NPs, but they may also activate the complement system, resulting in immunogenicity [50,51,52,53,54]. Nanocarriers that have a neutral charge and hydrophilic surface will more effectively utilize the EPR effect by accumulating in cancerous tissue. Coating hydrophilic polymers such as chitosan, polyethylenglycol (PEG), dextran, or polyvinylpyrrolidone (PVP) with nanocarriers has been one method for increasing their circulation duration [46]. However, nanocarriers must effectively overcome multiple transport hurdles in order to gain therapeutic effectiveness in treating GBM (including BBB). Active targeting may improve macromolecule selectivity even more than passive targeting. The use of cancer diagnostic proteins that bind to overexpressed cell surface proteins in particular cancer cells is one example of an active targeting method. macromolecular carriers that release anticancer medications into tumor tissue or tumor cells in response to internal or external stimuli are also included in this category. The tumor’s acidic pH, increased redox potential, and/or overexpressed proteins and enzymes all contribute to the release of drugs from the tumor’s cells. External stimuli, such as light, ultrasound, a magnetic field, and temperature, may also be used to release drugs and reach their molecular target in cancer cells [55]. The biosafety, prolonged drug release, greater solubility, improved bioactivity, BBB penetrability, and self-assembly of nanocarriers and nanotechnology-based drug delivery systems allow these colloidal-based particulate systems to bypass the BBB [43,56]. Many variables restrict the therapy choices for GB patients, and nanomedicine may be used to enhance the delivery, specificity, and effectiveness of both present and future treatments (Figure 5). 

## 4. Combination Therapy for Glioblastoma

### 4.1. Nano-Chemotherapies

Wnt signaling plays a crucial role in GBM, which affects apoptosis and autophagy by activating or inhibiting other pathways in the cell [57]. Curcumin (50 M), nanomicellar curcumin alone, and nanomicellar curcumin combination with TMZ were shown to dramatically reduce the invasion and migration of U-87 cells. Biomarkers of autophagy (Beclin 1 and LC3-I and -II) were found to be considerably elevated. As the levels of Bax protein decreased, those of apoptosis-related proteins Bcl-2 and caspase 8 increased. Genes related to the Wnt pathway (β-catenin, cyclin D1, Twist, and ZEB1) have drastically lower expression levels [58]. Apt-NPs, which were made from B19 aptamer (Apt)-conjugated polyamidoamine (PAMAM) G4C12 dendrimer nanoparticles (NPs) and used to deliver paclitaxel (PTX) and TMZ into U-87 stem cells, significantly reduced tumor growth in U-87 stem cells by inducing apoptosis and decreasing autophagy and multidrug resistance (MDR) gene expression [59]. YukinoriAkiyama and his colleagues found that, combining carmustine (BCNU) wafers and bevacizumab, newly diagnosed GBM patients treated with TMZ and radiation were shown to be safe. Patients with GBM responded better to the combination treatment than to normal therapy. This suggests that the combination treatment has a promising efficacy and side-effect profile [60]. An herbal polyphenolic molecule known as resveratrol (3,5,4′-trihydroxy-*trans*-stilbene) is found in red wine, peanuts, and soy. This herbal substance has some ability to destroy cancerous cells and enhance the tumor’s response to radiation or chemotherapy. The therapeutic effectiveness of resveratrol in GBM may be improved by its synergistic effects when combined with radiation and chemotherapy [61]. Nose-to-brain delivery of the conjugated NPs, which combine poly (D,L-lactic-*co*-glycolic acid) and chitosan nanoparticles with alpha-cyano-4-hydroxycinnamic acid (CHC) and cetuximab (CTX), was developed to treat GBM. EGRF activation was inhibited by CTX, which was shown to be covalently linked to NPs. When conjugated NPs were used in the chicken chorioallantoic membrane assay, there was a decrease in tumor size [62].

In a separate research project, scientists devised a liposomal delivery method that might be used to efficiently carry chemotherapy across the BBB to treat GBM. Tf-modified liposomes were used to target transferrin (Tf) and PFVYLI (PFV) cell-penetrating peptide (PFV) to boost the translocation of DOX and erlotinib across the BBB into U-87 tumor cells. In U-87 cells, brain endothelial cells, and glial cells, doxorubicin (DOX) and Erlo were efficiently absorbed. In addition, the apoptosis of U-87 cells was greatly increased by the use of dual-functionalized liposomes. Due to the increased BBB translocation of dual functionalized liposomes, around 52% tumor cell death was seen in in vitro brain tumor models employing the PLGA–chitosan scaffold-containing chemotherapy agents [63]. In order to achieve receptor-mediated transcytosis, a liposomal delivery method was produced that included a surface modified with transferrin (Tf) and a penetratin (Pen) cell-penetrating peptide. Loaded into liposomes, doxorubicin and the anti-glioblastoma drug erlotinib might more easily reach the cancerous tumor in the brain. There was a 15% increase in translocation across the coculture endothelium barrier when doxorubicin- and erlotinib-loaded Tf–Pen liposomes were delivered together to an in vitro brain tumor model, leading to tumor shrinkage and remission. Tf–Pen liposomes increased doxorubicin and erlotinib accumulation in the brains of mice by factors of 12 and 3.3, respectively, when compared to free medications. Tf–Pen liposomes regressed 90% of the tumor in mice brains, with a significant increase in median survival time (36 days) and no damage [64]. In comparison to their respective free drug formulations, codelivery of PTX- and methotrexate (MTX)-loaded PLGA NPs appear promising for the treatment of GBM [65].

### 4.2. Nano-Chemotherapy–Radiotherapy

Radiation therapy is one of the clinical therapies for GBM, and significant efforts have been undertaken to improve its effectiveness [66]. Radiation treatment, however, was ineffective because of the invasive tumor development of glioblastoma. Short-course radiation with the addition of TMZ in elderly patients with GBM resulted in longer survival than short-course radiotherapy on its own [67]. Humanized recombinant monoclonal antibody bevacizumab suppresses the proliferation and angiogenesis of vascular endothelial cells and has proven therapeutic effectiveness in the treatment of recurrent GBM [68]. One trial compared bevacizumab with RT/TMZ with RT/TMZ alone for newly diagnosed glioblastoma patients. There was no increase in overall survival (OS) but an improvement in progression-free survival (PFS). Additionally, the 6 month survival rate of rats given bevacizumab and radiotherapy/TMZ (RT/TMZ) was not improved. Adding bevacizumab to a radiotherapy/TMZ regimen may be a successful way to increase progression-free survival in patients [69]. An investigation showed that utilizing ionizing radiation (IR) in combination with the small-molecule inhibitor PLX3397 (a tyrosine-kinase inhibitor, CSF-1R inhibitor) was more effective than using IR alone in treating GBM intracranial xenograft mice [70]. Mice who received a combination of therapies had a considerably longer lifespan than those that received IR alone. Poly(lactic-*co*-glycolic acid) (PLGA) nanoparticles (PNPs) attached to chlorotoxin (CTX), Ag-PNP-CTX, may irradiate tumor cells and reduce the extracellular activity of MMP-2, according to a study. The use of CTX nanovectors in conjunction with radiation therapy may offer a potential treatment for GBM by lowering MMP-2 activity and targeting scattered GBM cells. The combination of CTX nanovectors and radiation may be an effective treatment for GBM. Because CTX targeting increases the accumulation of nanovector therapeutic cargo in GBM cells and suppresses them by about 50%, it has a synergistic effect [10,71].

DC101, an anti-VEGFR2 antibody, was shown in a study by Kozin et al. to reduce the needed dosage of radiotherapy for tumor growth suppression by 1.3–1.7 times by lowering hypoxia. Radiation treatment has been shown to normalize the vasculature, according to a number of studies. Apoptosis of ECs is increased in a dose-dependent manner by high-dose stereotactic body radiotherapy (SFRT), resulting in the normalization of tumor vasculature [72]. According to Lan et al., hypofractionated radiotherapy (HFRT) considerably increases OS when compared to traditional radiation treatment. It is still unclear exactly how HFRT works in terms of radiobiology. There were more arteries covered and more perfusion after ablative hypofractionated radiation, demonstrating that HFRT restored the tumor’s vasculature [73]. Antitumor drugs conjugated with gold nanoparticles (AuNPs) represent a potential and new treatment option. Using the low-density growth test and irradiation, the antitumoral efficacy of AuNP-SI306 was investigated in vitro in the GBM model. In combination with radiotherapy (RT), the combination of AuNPs and SI306 was more effective in inhibiting tumor cell growth than the combination of AuNPs and free SI306 [74]. U87-MG human GBM xenografts in nude mice were treated with radiation and chemotherapy using pH-sensitive polymersomes, which resulted in significantly longer lifetimes for the xenografts. The BBB-crossing capacity of Au-DOX@PO-ANG is excellent, allowing it to effectively target tumors. The pH sensitivity of this delivery system and its capacity to adapt to the tumor microenvironment make it ideal for this application. The combination of gold nanoparticles with doxorubicin is a novel medication. The radiotherapy (RT) effect on GBM is improved with this sort of complicated medication. Tumor volume was significantly reduced in mice treated with Au-DOX@PO-ANG NPs [75] (Table 1).

A potential noninvasive cancer treatment technique is the combination of photothermal therapy (PTT) with photodynamic therapy (PDT). As a result of light absorption, photothermal agents were used to create heat and destroy cancerous cells; as in the previous treatment, reactive oxygen species (ROS), free radicals, or peroxides were created to promote cell death [10]. Because of its selective therapeutic benefits, chemo-photothermal therapy (CPT) for cancer treatment is receiving increased interest. Fe_3_O_4_ magnetic nanoparticles drug-loaded with CPT have been shown to have anticancer effects on U-87 MG human GBM cells. Preparation of anticancer drug-loaded Fe_3_O_4_ MNPs by loading TMZ and indocyanine green (ICG) was carried out, and the samples were then examined using various techniques such as X-ray, UV–Vis spectroscopy and drug-loading capacity [76]. U-87 MG GBM cells died following MNP injection after being treated with near-infrared (NIR) light irradiation, resulting in the generation of reactive oxygen species (ROS). Researchers found that irradiating U-87 MG GBM cells with NIR laser-irradiated Fe_3_O_4_-TMZ-ICG MNPs significantly increased anticancer effects on Bcl-2-associated X protein, Bcl-2, cytochrome c, caspase-3, Fas-associated via the death domain, and caspase-8 genes, as shown by Western blot analysis and reverse transcription quantitative polymerase chain reaction. Using CPT, Fe_3_O_4_-TMZ-ICG MNPs may be a viable treatment option for patients with brain cancer [76]. Doxorubicin–curcumin–amino acid-based composite microbowls (CMBs) were created in 2021 by Chibh and coworkers using a miniaturized fluid flow-based self-assembly method. Dual chemo-photodynamic treatment was applied using CMBs on two- and three-dimensional (2D) spheroids of C6 glioma cells. CMBs with asymmetric holes showed promise as a combinatorial drug carrier for cancer treatment that might deliver chemo- and phototherapy at the same time [77]. PTT may be employed as a standalone therapy, guided by multimodal imaging, or used in combination with existing medicines for the treatment of cancer metastases, as shown by Zou et al. In several preclinical animal tests, a variety of photothermal nanotherapeutics (PTNs) have shown promising therapeutic effectiveness against metastatic cancer. PTT or a combination of PTT and other therapies may be an important and promising treatment option for cancer metastases [78].

By coating citrate-coated magnetic nanoparticles on rGO, researchers created an rGO-based magnetic nanocomposite (CMNP). To generate PEGylated mrGO for conjugation with gastrin-releasing peptide receptor (GRPR), phospholipid–polyethylene glycol was used to modify magnetic rGO (mrGOG). To transfer the anticancer medicine doxorubicin (DOX) into the endosome, the drug was coupled to mrGO (mrGOG) through π–π stacking interactions. It was shown that intravenous treatment of mrGOG/DOX under magnetic guidance reduced tumor growth and increased animal survival compared to groups that received free DOX or did not get magnetic guidance when using the U-87 tumor xenograft model developed in naked mice [79]. The anticancer effectiveness was greatly improved by increasing cell death and reducing cell proliferation when combined with a 5 min NIR laser therapy. Gold–silver nanotriangles (AuAgNTrs) that were stabilized by polyethylene glycol (PEG) were synthesized and used in photothermal treatment. Using the U-87 GBM cell model, a cell viability experiment was carried out. After just 10 min of laser irradiation at a power P = 3 W/cm^2^ that was shown to be nontoxic to the control cells, the excellent photothermal performance of AuAgNTrs was proven in suspension and in vitro [80]. Cell viability decreased by >80% after that time. The anti-EphA3-modified TMZ@GNPs (anti-EphA3-TMZ@GNPs) were synthesized for chemical and auxiliary plasma photothermal therapy (PPTT) in order to solve the issue of glioma resistance to TMZ, and to enhance GBM therapeutic benefits. TMZ@GNPs were used to treat GBM. In the anti-EphA3-TMZ@GNP-treated group, cytotoxicity and apoptosis were considerably greater than in the GNP and non-photothermal groups. Reversing drug resistance was shown by Western blot analysis, which indicated that the GNP–PPTT-mediated tumor cell death resulted in an increase in the production of antiapoptotic signaling molecules and cell-cycle inhibitors. After photothermal therapy, the anti-EphA3-TMZ@GNPs group survived 46 days longer (1.64-fold) than the TMZ group in the subcutaneous GBM model of nude mice [81]. For chemotherapy, a derivative of dicysteamine-modified hypocrellin (DCHB; a natural-origin photosensitizer) with a singlet oxygen quantum yield of 0.51 was used, together with a cyclic peptide (cRGD) as a targeting unit against GBM, to construct a multifunctional phototheranostic agent. As a result of the DCHB and TMZ-C18 assembly, the cRGD-decorated DTRGD NPs exhibited broad near-infrared absorption (peaking at 703 nm), NIR emission (peaking at 720 nm), strong photostability, a high photothermal conversion efficiency (peaking at 33%), and effective degradation of the TMZ-C18 compound. DTRGD NPs, on the other hand, may cross the BBB and target tumors directly. DTRGD NP-treated U-87MG tumor mice revealed that the targeted chemo/photodynamic/photothermal synergistic treatment may be accomplished with almost little harm [82].

uPAR, a plasminogen activator receptor of the urokinase type, is overexpressed in a variety of tumor species [83]. Indocyanine green (ICG)-conjugated peptide AE105, which targets the uPAR, has shown significant promise for fluorescence-guided surgery. During PTT, the photothermal abilities of ICG-Glu-Glu-AE105 led to tumor death and prolonged survival. Studies showed that apolipoprotein E peptide (ApoE), which targets the brain, grafted onto these nanoparticles, ApoE-Ph NPs, greatly increased PTT efficiency and the survival of mice with orthotopic GBM after mild irradiation (0.5 W·cm^−2^) [84]. It was shown that BK@AIE NPs, bradykinin aggregation-induced-emission nanoparticles, had a high photothermal conversion efficiency under 980 nm NIR laser irradiation, making them ideal for treating deep-seated malignancies. Tumor development can be significantly suppressed to lengthen the life span of mice following spatiotemporal PTT. Tissue necrosis factor and tumor-associated antigens may be eliminated and released by NIR irradiation. It was shown that the PTT treatment of GBM-bearing mice stimulated natural killer cells, CD3^+^ T cells, CD8^+^ T cells, and M1 macrophages in the GBM region, improving the therapeutic efficiency. BK@AIE NPs with NIR assistance were shown to be a potential technique for improving GBM clearance and activating local brain immune privilege in [85]. The hypoxic parts of tumors may be reached by macrophages that can penetrate blood vessel barriers. It has become a new trend to use macrophages as a medication delivery mechanism for tumor targeting. A photothermal agent, gold nanorods (GNRs), was effectively modified to boost cellular absorption and biocompatibility using monocyte chemoattractant protein-1 (MCP-1) and an iron-based metal–organic framework (MIL-100(Fe)). A xenograft model of U251 MG cells in nude mice was used to show the photothermal activity of MCP-1 and GNR@MIL-100 (Fe). After laser therapy, the tumor volume remained under 100 mm^3^ even after growth was reduced by further NIR treatment. Antitumor effectiveness of MCP-1/GNR@MIL-100 (Fe) coupled with laser therapy was shown by tumor histology, survival, and bioluminescence imaging [86]. Novel NPs for GBM PTT have been described in many articles; however, no clinical research has yet demonstrated their utility in GBM patients. Only a few researchers conducted in vitro analyses of GBM PTT, while others were able to overcome some of the fundamental problems of GBM PTT in vivo [87] (Table 2 and Table 3).

### 4.3. Nano-Chemotherapy–Immunotherapy

Oncology immunotherapy has garnered significant interest in the last several decades. There is a possibility that it might engage the body’s immune system and produce particular immunological responses to eradicate the tumor cells [137,138]. EGFRvIII, a tumor-specific epitope expressed in GBM, the most frequent and deadly primary malignant neoplasm of the brain, is the target of the peptide vaccine rindopepimut (CDX-110) [139]. As part of a multi-immunotherapy strategy, rindopepimut exhibited considerable therapeutic benefit and effectiveness in clinical studies. In the tumor microenvironment (TME), tumor-associated macrophages (TAMs) play a critical role. A shift from the protumor M2 (TAM2) to antitumor M1 (TAM1) phenotype lifts the immunosuppressive restrictions and enhances chemotherapy effectiveness [140]. It was shown that chemotherapy with macrophage-directed immunotherapy resulted in an improved therapeutic outcome. DOX@MSN-SS-iRGD&1MT nanoparticles were designed to simultaneously administer doxorubicin (DOX) and an immune checkpoint inhibitor (1-methyltryptophan, 1MT) into orthotopic glioma, where they showed promising results. To create the nanoparticle, silica nanoparticles loaded with DOX were coupled with Asp–Glu–Val–Asp (DEVD)-linked 1MT and then modified with iRGD, as shown in the figure. It was shown that these nanoparticles may pass across the BBB and reach the tumor site, where they significantly increase medication accumulation in orthotopic brain tumors while having common adverse effects [141]. Due to the active targeting of iRGD, the nanoparticles successfully crossed the BBB and boosted drug accumulation in orthotopic brain tumors [141]. DAMP emission from nano-DOX was shown to be more potent than that from doxorubicin. DTX-sHDL-CpG nanodisc administration into the tumor mass resulted in tumor regression and antitumor CD8^+^ T cell responses in the brain tumor microenvironment (TME), according to Kadiyala and her colleagues. In addition, 80% of GBM-bearing mice had tumor shrinkage and long-term survival after DTX-sHDL-CpG therapy combined with radiation (IR), which is the gold standard of care for GBM. For the treatment of GBM, the findings showed that nanodiscs in conjunction with IR led to tumor reduction, long-term survival, and immunological memory [142].

The tumor immune microenvironment may be altered, and chemotherapy’s efficacy may be improved by using RNAi-based immunomodulation, according to a study by Qiao et al. This system (Angiopep LipoPCB (TMZ + BAP/siTGF-β), ALBTA) was developed for the treatment of intracranial GBM with dual targeting and ROS response. Strong siRNA condensation, excellent drug loading efficiency, and good serum stability are among the properties of traceable nanoparticles. Through receptor-mediated transcytosis, they can penetrate the BBB and effectively target GBM cells. ALBTA’s zwitterionic lipid (distearoyl phos phoethanol-amine-polycarboxybetaine lipid) boosts TMZ’s cytotoxicity and improves the gene silencing efficacy of siTGF-β by promoting endosomal/lysosomal escape. ALBTA significantly improves the improved immunosuppressive microenvironment of glioma-bearing mice [143]. To change the immunological milieu of GBM and enhance the efficiency of TMZ, siRNA against tumor growth factor (siTGF-β) was used. To deliver these two medications in a regulated way, we first selected an ROS-responsive poly[(2-acryloyl)ethyl(*p*-boronic acid benzyl) diethylammonium bromide] (BAP) to join with siTGF-β and trigger its release into the cytoplasm of tumor cells. Second, a zwitterionic lipid distearoyl phosphoethanol-aminepolycarboxybetaine (DSPE PCB)-based envelope (ZLE) was chosen to increase TMZ and BAP/siTGF-β (AN@ siTGF-β) transport into the cytoplasm. In cerebral glioma mice, the core–shell structural nanoparticles (ALBTA) considerably alleviated the immunosuppressive milieu in vivo and raised the median survival time from 19 to 36 days without evident systemic harm. Combining immunotherapy and chemotherapy through nanotechnology significantly increased the susceptibility of GBM cells to chemotherapeutic agents and regulated the tumor microenvironment [10,143].

Galstyan et al. investigated the potential of targeted nanoscale immunoconjugates (NICs) produced utilizing poly(l-malic acid), a naturally occurring polymeric scaffold covalently labeled with a-CTLA-4 or a-PD-1, by examining their systemic distribution across the blood–brain barrier (BBB). The local anticancer immune response in the brain demonstrated that the checkpoint blockade drug was delivered across the BBB to the tumor location, implying the induction of a systemic and local immune response in glioblastoma therapy [144]. Another intriguing study on immune nanoconverters encapsulating a resiquimod- and doxorubicin-loaded scaffold demonstrated the polarization of immunosuppressive tumor-associated macrophages (TAMs) and myeloid-derived suppressor cells into tumoricidal APCs, as well as in situ vaccination via in vivo mechanisms for the activation of neoantigen-specific T cells [145]. Chemotherapy and PDT (photodynamic treatment) in conjunction with ICB (immune checkpoint blockade) are also commonly used in many malignancies. Combining local chemotherapy and anti-PD-1 therapy may improve antitumor immune responses and prolong overall survival in glioblastoma treatment. Notably, the chemo- and immunotherapy sequence is essential for defining the anti-PD-1 antibody’s activity [146,147].

Gold nanoparticles (AuNPs) and outer membrane vesicles (OMVs) from *E. coli* were successfully used to construct an Au–OMV complex. For both subcutaneous G261 tumor-bearing C57BL/6 mice and in situ (brain) tumor-bearing C57BL/6 mice, the combination of radiation and Au–OMV induced radiosensitizing and immunomodulatory effects that effectively reduced tumor development. In situ tumor-bearing mice treated with Au–OMV and radiation had a longer survival time. The treatment’s mechanisms of success were examined. Au–OMV and radiation enhanced intracellular ROS in G261 glioma cells [148].

Furthermore, it was shown that the vitality of G261 glioma cells was linked to macrophage chemotaxis and the generation of TNF-α in the presence of RAW 264.7 macrophages [148]. Increasing the immunogenicity of GBM cells (GC) is a promising strategy for overcoming the immunosuppression associated with GBM. An immunosuppressive microenvironment in GBM was efficiently altered by nanodiamonds containing doxorubicin (Nano-DOX), which was shown to stimulate the GC’s immunogenicity and start anti-GBM immune responses [149]. Researchers discovered that Nano-DOX induced GC to release antigens and damage-associated molecular patterns (DAMPs), which operate as potent adjuvants, rather than apoptosis. As a consequence, dendritic cells (DC) were more activated. In Nano-DOX-treated GC, an increase in autophagosome release was noted. However, it was shown to be a minor source of antigen donation. Nano-DOX-induced GC antigen donation and DAMP emission were decreased by blocking autophagy in GC, although DC activation was also effectively suppressed by Nano-DOX-treated GC. These data imply that Nano-DOX increases GC immunogenicity primarily via activation of autophagy. By leveraging autophagy in cancer cells, nanotechnology may be used to alter the GBM immune microenvironment therapeutically [150,151]. Immunosuppression and treatment resistance in GBM are mainly attributed to tumor-associated myeloid cells (TAMCs). Since up to 50% of the brain tumor mass is composed of TAMCs, it is imperative that a treatment approach for targeting TAMCs in GBM be developed. These studies showed that an LNP platform can recognize highly expressed programmed death-ligand 1 (PD-L1) in tumor-associated macrophage cells, which allows it to selectively target and deliver drugs to tumors in mice and humans. Dinaciclib-encapsulated lipid nanoparticles (LNPs) effectively eliminated TAMCs from tumors and dramatically improved the survival of mice in glioma models (GL261 and CT2A) when used in conjunction with radiation treatment. This nanomedicine platform has the potential to revolutionize the treatment of GBM and speed up its adoption in clinical practice [152] (Table 4).

## 5. Conclusions and Future Perspective

The poor prognosis of GBMs is well known, and current treatment approaches have achieved little to increase overall survival or survival without progression [41]. As GBM treatment research progresses, nanotechnology has emerged as a promising approach. Research on the long-term effects of nanomaterials on human health must be conducted systematically and comprehensively. The therapeutic application of nanomaterials will need significant improvements in both aspects, including reducing dosages and reducing exposure duration without compromising the particular intratumoral accumulation of the nanoparticles [68]. Clinical studies involving nanomedicines are presently taking place; nonetheless, liposomes and polymeric nanoparticles remain the primary emphasis. There is a pressing need to test numerous developing and flexible materials, such as exosomes, hitchhiked nanocarriers, and porous materials (silica, silicon, and MOFs), in treating brain illnesses, such as glioma. These materials can overcome many biological obstacles, which means they have great promise for use in the treatment of GBM, particularly in the delivery of immunotherapeutic drugs. Their clinical transferability, including repeatability, scaling, biocompatibility, toxicity, and patient access, is essential. Novel materials being studied need new models for in vitro and in vivo studies to better understand the destiny of this nanomedicine, especially in the context of more contemporary materials. Extensive data may be obtained from in vitro models, such as patient-generated spheroids and 3D-printed GBM on-chip models, when they are used to validate novel compounds, devices, or drug delivery methods. Physical, chemical, and biological interventions should be combined to treat GBM and improve patient quality of life, while simultaneously overcoming the disease’s inherent heterogeneity and resistance to single treatments [153]. Future research should examine issues such as safety, biocompatibility, accessibility, and toxicity, among other things. These nano-combinations are also likely to be more expensive than the sum of the prices of the individual drugs. If cancer therapies are to be shown effective, they must be tested in a range of animal models, as well as potential clinical trials on human subjects. In addition, further research on the timing of therapy techniques must be conducted. For example, the effectiveness of immunotherapy is clearly affected by the time of chemotherapy treatment in combination. There is always a need for such research into the complex tumor microenvironment. Since the GBM’s distinct targets and acceptable stimulus responses must be exploited, the various agents may be released in appropriate locations, and adequate therapeutic agents can be delivered to GBM cells. In addition, the FDA has not approved the majority of the biomaterials used in trials, making clinical translation even more challenging [10,154,155].

## Figures and Tables

**Figure 1 pharmaceutics-14-01697-f001:**
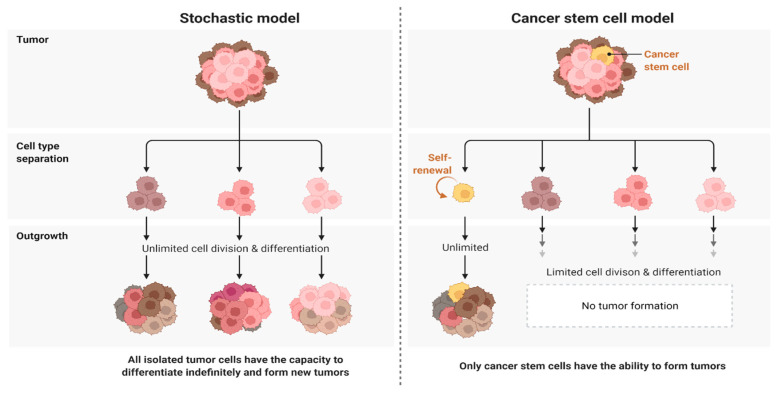
Stochastic vs. cancer stem-cell hypothesis. The differentiation hypothesis states that all cells can become cancerous, but only some of them will contribute to tumor formation in response to a specific set of stimuli. According to the stem-cell hypothesis, only a small percentage of cells, known as cancer stem cells, can self-renew, initiate, and regrow a tumor. Created with BioRender.com.

**Figure 2 pharmaceutics-14-01697-f002:**
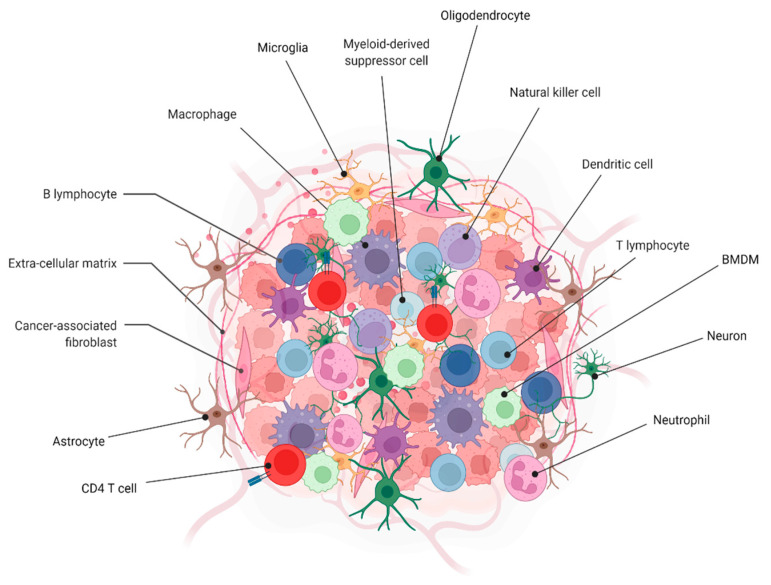
Highlighting the diversity of tumor–host cell interactions in GBM. Created with BioRender.com.

**Figure 3 pharmaceutics-14-01697-f003:**
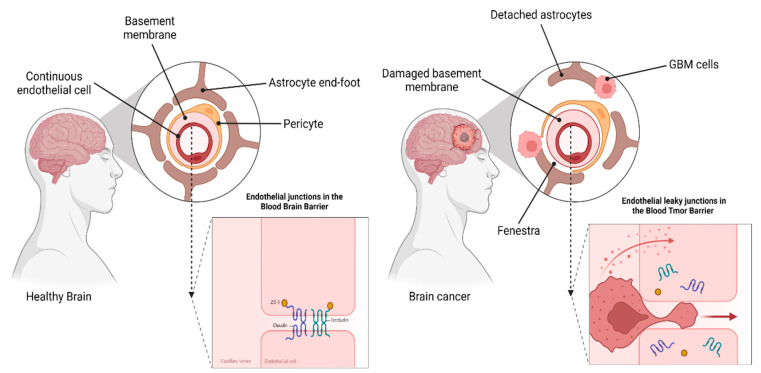
Schematic representation of the BBB vs. BTB in GBM. Healthy brain: The BBB is made up of a variety of cell types and ECM molecules working very closely. The neurovascular unit (NVU) contains endothelial cells, basal lamina cells, pericytes, and astrocyte end-feet that are highly specialized and polarized and wrap the micro-vessel walls in order to communicate with neurons. Brain cancer: Damage to the NVU and endothelial permeability occurs in the BTB owing to astrocyte displacement, neurovascular decoupling, changed pericyte populations, alterations in tight junctions, and changes in endothelial cell (EC) transcytosis mechanisms [36,40]. Created with BioRender.com.

**Figure 4 pharmaceutics-14-01697-f004:**
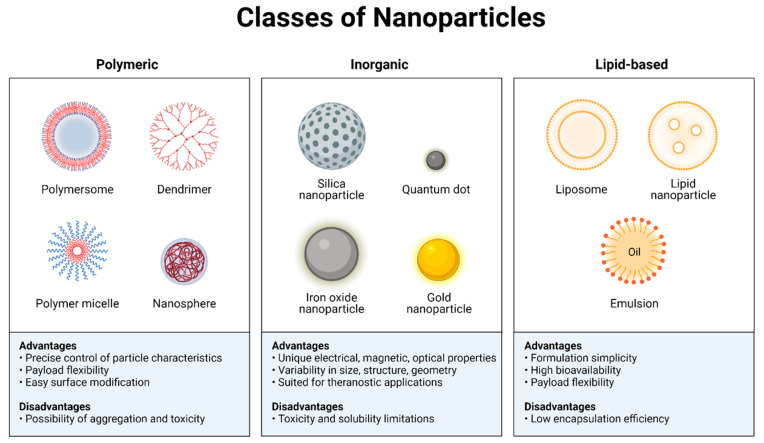
Different types of nanoparticles used in drug delivery systems for cancer therapy. Created with BioRender.com.

**Figure 5 pharmaceutics-14-01697-f005:**
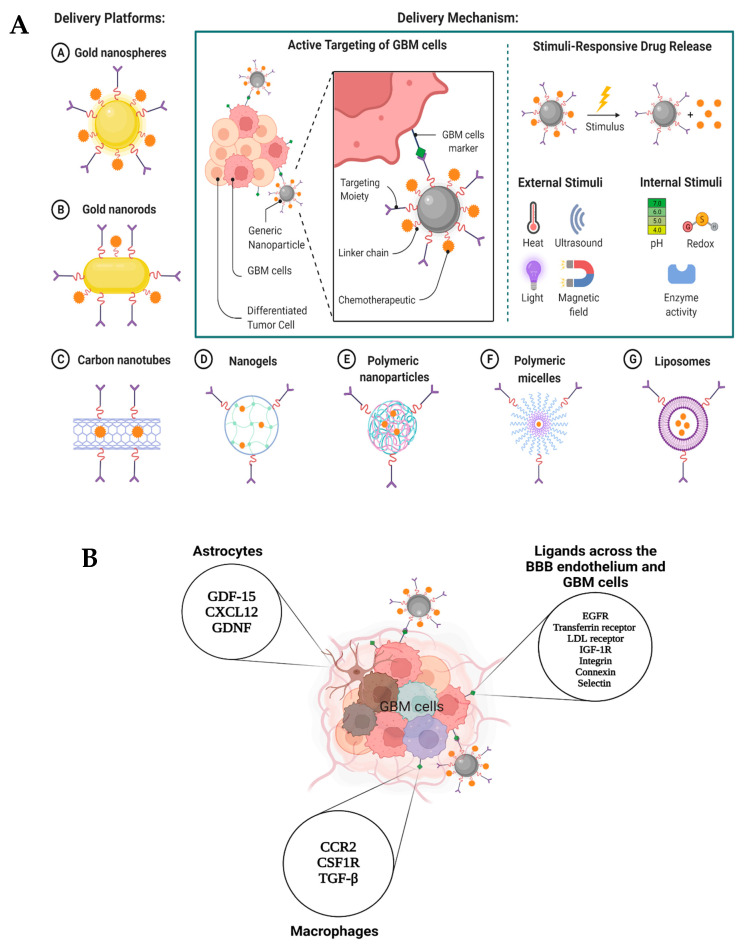
Nanoparticle-mediated targeted drug delivery to GBM cells. (**A**) For the elimination of GBM cells, surface markers unique to GBM cells serve as possible therapeutic targets. The conjugation of targeting moieties to drug-carrying nanoparticles is one way to accomplish active targeting. These molecules are capable of binding to GBM cells markers. A variety of external and internal cues may be used to initiate the release of drugs after they have become bound. As seen in the picture, a variety of nanoparticle-based drug delivery platforms have been created using these methodologies (A–G). (**B**) Therapeutic targets in BBB endothelium and GBM cells [46]. Created with BioRender.com.

**Table 1 pharmaceutics-14-01697-t001:** The studies reporting nano-chemotherapy–radiotherapy in GBM.

Molecules	Suggested Mechanism of Action	Result
TMZ + RT	-	Longer survival
Bevacizumab + TMZ + RT	Suppresses the proliferation and angiogenesis of vascular endothelial cells	Therapeutic effectiveness
IR + PLX3397	-	Longer lifespan
Ag-PNP-CTX	Reduce the extracellular activity of MMP-2	-
CTX nanovector + RT	Increases the accumulation of nanovector therapeutic cargo in GBM cells	Synergistic effect suppresses by about 50%
DC101 + RT	Lowering hypoxia	Tumor growth suppression
AuNPs-SI306 + RT	-	Inhibition of tumor cell growth
Au-DOX@PO-ANG NPs + RT	Increase BBB-crossing capacity	Reduction in tumor volume

Abbreviations: TMZ (temozolomide); RT (radiotherapy).

**Table 2 pharmaceutics-14-01697-t002:** In vitro cellular and in vivo preclinical studies reporting NPs for GBM PTT [87].

Photoabsorbing Agent	PTT Laser and Treatment Conditions			Preclinical Model	Model	Reference
	Power (W/cm^2^)	Exposure Time (min)	Administration Regimen/Route			
RVG29-SiO_2_-PEG-AuNR	1.5	5	iv	N2a neuroblastoma		[88]
4Cu-RGD-Au NR	1	10	iv	U87 MG		[89]
AuNS	4	3	iv	U373 GBM		[90]
RGD-AuNSt	1	10	iv, multiple	U87 MG		[91]
PPDI-PEG-Au NP	0.3	5	iv	U87 MG		[92]
rGONM-PEG-Cy7-RGD	0.1	7	iv	U87 MG		[93]
PNG-RGD	2.5	5	it	U87 MG		[94]
C225-EPI-PEG-NGO	2	2	iv	U87 MG		[95]
rGO-AuNRVe-DOX	0.25	5	iv	U87 MG		[96]
pDNA-loaded AuNR-Fe_3_O_4_NS	2	5	it, multiple	U87 MG		[97]
C225-Au-MNP	0.3	30	pt, multiple	C6		[98]
I-RGD-PEG-MNP	0.5	5	iv, multiple	U251		[99]
ANG-Au-PLGA-DTX NPs	1.5	1.5	iv, multiple	U87 MG		[100]
UCNP-PEG-ICG-TOS-RGD	0.5	5	iv, multiple	U87 MG		[101]
ASQ-DOX-PGEA2/p53 nanohybrids	2	5	it, multiple	C6		[102,103]
I RGD-CR780-PEG NPs	0.5	10	iv	U87 MG		[104]
melittin/ICG peptide nanofiber hydrogel	2	8	it	C6		[105]
CuS–Fn NCs	0.8	5	iv	U87 MG		[106]
PPyHMs	0.64	10	it	U87 MG		[107]
holo-Tf-ICG	0.8	5	iv	U87 MG		[108]
CPNP	0.8	5	iv	U87 MG		[109]
Ma-AuNS	N/A	10	it	C6		[110]
cRGD-PEG-HAuNS	16	3	iv	U87 MG-Luc		[111]
VEGF-AuNS	3	6	iv	U373 GBM		[112]
Tf-TPGD	2.5	5	iv, multiple	C6		[113]
HCCD	1	5	iv, multiple	U87 MG		[114]
OMCN–PEG–Pep22/doxycycline	N/A	5	iv, multiple	C6		[115]
ANG-IMNPs	0.21	3	iv	ALTS1C1 astrocytoma		[20]
cRGD-CPNP	0.8	5	iv	U87 MG-Luc		[116]
BLIPO-ICG	1	5	iv	C6-Luc		[117]
AuNR	1.2 W *			1321N1 human astrocytoma	2D	[118]
Nes-AuNR	0.5			X01 GBM, X01 GBM-BMP	2D, 3D	[119]
AuNS	80			U373, U87 MG	2D	[120]
Ma-AuNS	2, 7, 14, or 28			ACBT human glioma	2D, 3D	[121,122]
AuNSt@probe	2			U87 MG	2D	[123]
AuNSt-ICG-BSA	1			U87 MG	2D	[124]
CPT-GNC	76 **			42 MG-BA	2D	[125]
r1-AuSiO2 NP	4			U87 MG	2D	[126]
TiN NP	4.4			U87 MG	2D, 3D	[127]
nano-rGO-RGD	15.3			U87 MG	2D	[128]
nanoGO-Tf-FITC	7.5			U251 glioma	2D	[129]
PVP-G	2			U251 glioma	2D	[130]
DOX-GMS-PI	6			U251 glioma	2D	[131]
IUdR-PLGA-NGO	2			U87 MG	2D	[132]
MWCNTS	3			U87 MG, U373, D54	2D, 3D	[133]
PDA-ICG-NDs	2 W *			U-118 MG	2D	[134]
ICG-PL-PEG	0.75 to 3.25			U87 MG	2D	[135]
FA-Au-NP	8.5			C6 glioma	2D	[136]

Abbrivations: PTT, photothermal therapy; it, intratumoral; iv, intravenous; pt, peritumoral; NPs, nanoparticles; RVG29, 29-residue peptide rabies virus glycoprotein; PEG, polyethylene glycol; AuNR, gold nanorods; RGD, arginine–glycine–aspartic acid peptide; AuNS, Au@SiO_2_ nanoshells; AuNSt, gold nanostars; PPDI, poly(perylene diimide); rGO, reduced graphene oxide; rGONM, rGO nanomesh; Cy7, cyanine 7; PNG, porphyrin-immobilized nanographene oxide; C225, cetuximab; EPI, epirubicin; rGO-AuNRVe, hybrid reduced graphene oxide-loaded ultrasmall gold nanorod plasmonic vesicles; DOX, doxorubicin; pDNA, plasmid DNA; Fe_3_O_4_NS, Fe_3_O_4_ nanospheres; Au-MNPs, core–shell Fe_3_O_4_@Au magnetic nanoparticles; MNPs, Fe@Fe_3_O_4_ magnetic nanoparticles; ANG, angiopep-2 peptide; PLGA, poly(lactide-*co*-glycolide); DTX, docetaxel; UCNP, cesium-based upconversion nanoparticles; ICG, indocyanine green; TOS, alpha-tocopheryl succinate; ASQ-PGEA2, multifunctional heteronanoparticles comprising Au NRs, mesoporous silica, quantum dots and two-armed ethanolamine-modified poly(glycidyl methacrylate) with cyclodextrin cores; CP NPs, donor/acceptor conjugated polymer nanoparticles; CR780, croconaine; CuS–Fn NCs, ultrasmall copper sulfide NPs loaded inside the cavity of ferritin nanocages; PPyHMs, polypyrrole hollow microspheres; holo-Tf, holo-transferrin nano assemblies. N/A, not applicable; AuNS, Au@SiO_2_ nanoshells; VEGF, vascular endothelial growth factor; Ma-NS, AuNS-loaded macrophages; cRGD, cyclic arginine–glycine–aspartic acid peptide; PEG, polyethylene glycol; HAuNS, hollow gold nanospheres; Tf, transferrin; TPGD, nanoscale graphene oxide loaded with doxorubicin; HCCD, highly crystalline carbon nanodots; OMCN, oxidized nanocrystalline mesoporous carbon particles; Pep22, Pep22 polypeptide; ANG, angiopep-2 peptide; IMNP, hybrid nano-assemblies loaded with IR-780 (PTT agent) and mTHPC (PTD agent); CP NPs, donor/acceptor conjugated polymer nanoparticles; BLIPO-ICG, biomimetic proteolipid nanoparticles; ICG, indocyanine green; * power density; ** pulsed laser average power density; AuNRs, Au nanorods; Nes, nestin; PEG, polyethylene glycol; AuNS, Au@SiO_2_ nanoshells; Ma, macrophages; AuNSt, Au nanostars; AuNSt@probe, AuNSt conjugated with Atto 655 dye, Asp–Glu–Val–Asp peptide, and a folic acid; ICG, indocyanine green; BSA, bovine serum albumin; CPT, camptothecin; GNC, mesoporous silica-coated Au nanocluters; NP, nanoparticles; AuSiO_2_, silica NPs with a gold core; TiN, titanium nitride; nano-rGO, nanosized reduced graphene oxide; RGD, arginine–glycine–aspartic acid peptide; Tf, transferrin; FITC, fluorescein isothiocyanate; PVP-G, polyinylpyrrolidone-coated graphene sheets; GMS-PI, interleukin 13 peptide modified mesoporous silica-coated graphene nanosheet; DOX, doxorubicin; IUdR, 5-iodo-2-deoxyuridine; PLGA, poly lactic-*co*-glycolic acid; NGO, nanographene oxide functionalized with poly lactic-*co*-glycolic acid; MWCNTS, phospholipid–poly(ethylene glycol)-coated multiwalled carbon nanotubes; PDA, polydopamine; NDs, nanodiamonds; PL-PEG, phospholipid–polyethylene glycol; FA, folic acid; Au-NPs, Au-decorated polymeric NPs. Reproduced with permission from [87].

**Table 3 pharmaceutics-14-01697-t003:** The studies reporting nano-chemotherapy–phototherapy in GBM.

Molecules	Suggested Mechanism of Action	Results
Fe_3_O_4_-TMZ-ICG MNPs	Effects on Bcl-2-associated X protein, Bcl-2, cytochrome c, caspase-3, Fas-associated via the death domain, and caspase-8 genes	Increased anticancer effects
Doxorubicin–curcumin–amino acid (CMBs)	Drug carrier for cancer treatment	Treatment using CMBs on two- and three-dimensional (2D) spheroids of C6 glioma cells
mrGOG-DOX	DOX coupled to mrGO (mrGOG) through the binding of π-π stacking	Tumor reduction, long-term survival
Gold-silver nanotri-angles (AuAgNTrs)	Becomes nontoxic to cells	Cell viability decreased by >80%
anti-EphA3-TMZ@GNPs	Boosts TMZ’s cytotoxicity and apoptosis	Increase in the production of antiapoptotic signaling molecules and cell-cycle inhibitors
DCHB-TMZ-C18	Cross the BBB and target tumors directly	Targeted chemo/photodynamic/photothermal synergistic treatment with little harm
ICG-Glu-Glu-AE105	Targeting plasminogen activator receptor (uPAR)	Tumor death and prolonged survival
ApoE-Ph NPs	Increases PTT efficiency	Increases the survival of mice with orthotopic GBM
MCP-1/GNR@MIL-100 (Fe)	Boost cellular absorption and biocompatibility	Antitumor effectiveness
BK@AIE NPs-NIR	Removal and release of tissue necrosis factor and tumor-associated antigens by NIR irradiation	Improving GBM clearance and activating local brain immune privilege

**Table 4 pharmaceutics-14-01697-t004:** The studies reporting nano-chemotherapy–immunotherpy in GBM.

Molecules	Suggested Mechanism of Action	Results
Rindopepimut (CDX-110)	EGFRvIII	Multi-immunotherapy/enhances chemotherapy effectiveness
Doxorubicin + (1-methyltryptophan, 1MT)	Immune checkpoint inhibitor	Drug accumulation in orthotopic brain tumors
DTX-sHDL-CpG nanodisc + IR	Antitumor CD8^+^ T-cell responses in the brain tumor microenvironment (TME)	Tumor reduction, long-term survival
Angiopep LipoPCB (TMZ + BAP/siTGF-β), ALBTA	Chemotherapy + RNAi-based immunomodulation	Boosts TMZ’s cytotoxicity/improves gene silencing efficacy of siTGF-β ALBTA
ALBTA’s zwitterionic lipid (distearoyl phos phoethanol-amine-polycarboxybetaine lipid + TMZ)	Boosts TMZ’s cytotoxicity and improves gene silencing efficacy of siTGF-β by promoting endosomal/lysosomal escape	Increases the susceptibility of GBM cells to chemotherapeutic agents/regulated the tumor microenvironment
Immunoconjugates (NICs) + a-CTLA-4 or a-PD-1	Checkpoint blockade drug delivered across BBB to the tumor location	Induction of a systemic and local immune response in glioblastoma therapy
Resiquimod + doxorubicin	Activation of neoantigen-specific T cells	Polarization of immunosuppressive tumor-associated macrophages (TAMs)
Chemotherapy + anti-PD-1	Improves antitumor immune responses	Prolong overall survival in glioblastoma treatment
AuNPs + OMVs-(Au–OMV)	Induces radiosensitizing and immunomodulatory effects	Reduced tumor development
Chemotaxis + TNF-α		
Immunosuppressive microenvironment + doxorubicin (Nano-DOX)	Increasing the immunogenicity of GBM cells (GC)	Initiation of anti-GBM immune responses
Nano-DOX + dendritic cells (DC)	Increases GC immunogenicity via activation of autophagy	Alteration of the GBM immune microenvironment

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
