# Peer review of "Nanotechnology-Based Combinatorial Anti-Glioblastoma Therapies: Moving from Terminal to Treatable"

_pharmaceutics, 2022, doi:10.3390/pharmaceutics14081697_

Round 1

Reviewer 1 Report

Amir Barzegar behrooz et al. reviewed recent progress of nanotechnology-based combination therapy for glioblastoma treatment. It is a hot and important topic in the field of drug delivery systems. Overall, it is well-organized and deserved to be published in pharmaceutics. But before publication, the authors should address the following concerns.

  1. First, the authors should add several other sections to make a more logically clear story. For example, i, Biological barriers for GBM delivery (BBB, BBTB...) should be discussed; ii, How nanotechnology can help to improve the delivery to GBM (passive targeting, active targeting....). Some useful information might be found here: https://doi.org/10.1021/jacs.0c09029
  2. The authors should explain why nanotechnology is needed for combination therapy (minimizing toxicity of small molecule-based combination therapy, unifying pharmacokinetics...).
  3. It is better to show several representative figures (not only tables).
  4. Conclusion is too broad. Please give your own thoughts for future development.
  5. Please check language. For example, 

    Table 1. Combination therapy for GBM treatment is being tested in clonical studies [40]. 'clonical' should be 'clinical'.

Author Response

Amir Barzegar behrooz et al. reviewed recent progress of nanotechnology-based combination therapy for glioblastoma treatment. It is a hot and important topic in the field of drug delivery systems. Overall, it is well-organized and deserved to be published in pharmaceutics. But before publication, the authors should address the following concerns.

Q1: First, the authors should add several other sections to make a more logically clear story. For example, i, biological barriers for GBM delivery (BBB, BBTB...) should be discussed; ii, how nanotechnology can help to improve the delivery to GBM (passive targeting, active targeting....).

A1: Thank you so much for this valuable comments. We added title and subtitles in the manuscript and completely covered the sections that was asked. Page 2-5 in red colours.

Q2: The authors should explain why nanotechnology is needed for combination therapy (minimizing toxicity of small molecule-based combination therapy, unifying pharmacokinetics...).

A2: Thanks for the comment. The role of nanotechnology was added in the manuscript. Page 5-8 in red colours.

Q3: It is better to show several representative figures (not only tables).

A3: Thanks for the comment. Five representative figures were added in the manuscript.  Page 3-8.

Q4: Conclusion is too broad. Please give your own thoughts for future development.

A4: Thanks for the comment. The future perspective of conclusion was re-written.  Page 20.

Reviewer 2 Report

In this review, the authors have disused about the combination therapies that are developed using nanotechnology-based delivery techniques, for the treatment of GBM. It is well written. However, the authors just discussed the previous works and their final outcomes, without discussing about the mechanism how one therapeutic modality affects the efficacy of other modality in the process of combination. So, the discussion part should be rewritten by providing the concept for combination, explanation to the key outcomes of few works with images, and tables containing previous works.

1. Table 1, correct the typo “clonical” to “clinical”.        

Author Response

In this review, the authors have discussed about the combination therapies that are developed using nanotechnology-based delivery techniques, for the treatment of GBM. It is well written. However, the authors just discussed the previous works and their final outcomes, without discussing about the mechanism how one therapeutic modality affects the efficacy of other modality in the process of combination. So, the discussion part should be rewritten by providing the concept for combination, explanation to the key outcomes of few works with images, and tables containing previous works.

  1. Table 1, correct the typo “clonical” to “clinical”.

Q1: However, the authors just discussed the previous works and their final outcomes, without discussing about the mechanism how one therapeutic modality affects the efficacy of other modality in the process of combination. So, the discussion part should be rewritten by providing the concept for combination, explanation to the key outcomes of few works with images, and tables containing previous works

A1: Thank you so much for these valuable comments. We added new sections in the manuscript to answer the parts that were asked. First, we added the “Biological Challenges of Glioblastoma Therapy: More than Meets the Eye” section. In this part, we elaborated on the common challans in glioblastoma therapy. Then, we added the section “Multimodality therapeutic approaches in glioblastoma.” In this part, we explained the role of nanotechnology and how it affects combination therapy. In the next step, table 1, table 2, and table 3 not only summarized essential outcomes of the previous work but also added a mechanism column to indicate the mechanism of combination therapy. In addition, five representative figures were added to the manuscript. Page 2-8

Round 2

Reviewer 1 Report

The authors addressed the concerns well.

Reviewer 2 Report

The revised manuscript is qualified for publication.